# Novel Bat Adenovirus Closely Related to Canine Adenoviruses Identified via Fecal Virome Surveillance of Bats in New Mexico, USA, 2020–2021

**DOI:** 10.3390/v17101349

**Published:** 2025-10-08

**Authors:** Taylor E. Weary, Lawrence H. Zhou, Lauren MacDonald, Daniel Ibañez IV, Chance Jaramillo, Christopher D. Dunn, Timothy F. Wright, Kathryn A. Hanley, Tony L. Goldberg, Teri J. Orr

**Affiliations:** 1Department of Pathobiological Sciences, University of Wisconsin-Madison, 1656 Linden Drive, Madison, WI 53706, USA; weary@wisc.edu (T.E.W.); cddunn2@wisc.edu (C.D.D.); 2Veterinary Services Unit, Wisconsin National Primate Research Center, 1220 Capitol Court, Madison, WI 53715, USA; 3Department of Biology, New Mexico State University, Las Cruces, NM 88003, USA; lzhou@nmsu.edu (L.H.Z.); laurenelmacdonald@gmail.com (L.M.); dannyiv@ku.edu (D.I.I.); cjaramillo181@gmail.com (C.J.); wright@nmsu.edu (T.F.W.)

**Keywords:** bats, adenovirus, SARS-CoV-2, COVID-19, spillover, virome, viral surveillance, metagenomic sequencing

## Abstract

Bats host a wide range of viruses, including several high-profile pathogens of humans and other animals. The COVID-19 pandemic raised the level of concern regarding the risk of spillover of bat-borne viruses to humans and, conversely, human-borne viruses to bats. From August 2020 to July 2021, we conducted viral surveillance on 254 bats from 10 species across urban, periurban, and rural environments in New Mexico, USA. We used a pan-coronavirus RT-PCR to assay rectal swabs and performed metagenomic sequencing on a representative subset of 14 rectal swabs and colon samples. No coronaviruses were detected by either RT-PCR or metagenomic sequencing. However, four novel viruses were identified: an adenovirus (proposed name lacepfus virus, LCPV), an adeno-associated virus (AAV), an astrovirus (AstV), and a genomovirus (GV). LCPV, detected in a big brown bat (*Eptesicus fuscus*), is more closely related to canine adenoviruses than to other bat adenoviruses, suggesting historical transmission between bats and dogs. All virus-positive bats were either juvenile or adult individuals captured in urban environments; none exhibited obvious clinical signs of disease. Our findings suggest limited or no circulation of enzootic coronaviruses or SARS-CoV-2 in southwestern U.S. bat populations during the study period. The discovery of a genetically distinct adenovirus related to canine adenoviruses highlights the potential for cross-species viral transmission and underscores the value of continued virome surveillance in animals living with and near humans.

## 1. Introduction

Bats are recognized as important reservoirs for many viruses significant to animal and human health. As the only mammals capable of sustained flight and possessing unique adaptations in both immunity and life histories, bats host an exceptionally high viral diversity, typically without displaying overt signs of disease [1]. These viruses include high-profile zoonotic agents such as rabies virus, filoviruses (Ebola and Marburg), paramyxoviruses (Nipah and Hendra), and multiple coronaviruses (SARS-CoV, MERS-CoV, and SARS-CoV-2) [2,3]. The frequent emergence of bat-associated viruses underscores the need to understand their ecology and epidemiology around the world, given that bats are found on every continent except Antarctica [4].

For approximately one year (August 2020–July 2021), we conducted a viral surveillance study of bats in the Southwest United States, a region where previous studies have detected a diversity of novel viruses in North American bat species [4,5,6,7,8,9]. Furthermore, because North American bat virus surveillance studies prior to the COVID-19 pandemic often relied on convenience sampling, we aimed to systematically sample bat species with various behavior and life history traits as well as throughout all seasons of the year. Our primary goal in this study was to surveil for coronaviruses, and our secondary goal was to fill important knowledge gaps about North American bat viromes to guide decision-making about human health as well as bat research and conservation activities. Combining data on demography and enzootic virome of target taxa can make bat research safer for both bat populations and people by identifying pertinent risks and best practices to mitigate them [10,11].

While we did not detect any coronavirus infections in the sampled bats, we found some bats infected with several other viruses, notably a novel adenovirus. Among the diverse viral families harbored by bats, adenoviruses (family: *Adenoviridae*) are relatively understudied, despite their global distribution and ability to infect a wide range of vertebrate hosts. Adenoviruses (AdVs) are non-enveloped, double-stranded DNA viruses that cause a variety of clinical manifestations in their hosts, ranging from asymptomatic infections to severe respiratory, gastrointestinal, hepatobiliary, and ocular diseases [12]. Although human AdVs have been studied extensively, our understanding of AdVs in wildlife, especially in bats, remains incomplete, despite evidence suggesting zoonotic transmission of some AdVs [13]. Recent viral surveillance studies have begun to uncover the diversity of AdVs in bat populations, suggesting the existence of numerous uncharacterized lineages with varied evolutionary trajectories [14,15,16,17,18,19]. In this study, we report the identification and molecular characterization of a novel AdV detected in a big brown bat (*Eptesicus fuscus*).

## 2. Materials and Methods

### 2.1. Study Site and Sample Collection

Between 3 August 2020, and 28 July 2021, insectivorous bats (n = 254) of at least 10 different species (*Antrozous pallidus*, *Corynorhinus townsendii*, *Eptesicus fuscus*, *Lasiurus cinereus*, *Myotis* species (*M. californicus*, *M. ciliolabrum*, *M. evotis*, *M. volans*, *M. yumanensis*, or uncharacterized), and *Tadarida brasiliensis*) were trapped in urban, periurban, and rural locations (Figure 1) around Doña Ana County, New Mexico (Table 1), under New Mexico Department of Game and Fish permit number 3782. All animals were trapped and handled in accordance with the American Society of Mammalogists guidelines for use of wild mammal species in research [20], and animal work was approved by the New Mexico State University Institutional Animal Care and Use Committee (2020-005). An exemption was granted by New Mexico Game and Fish from the 2020 embargo on bat research during the COVID-19 pandemic; all animal handlers wore N95 masks, face shields, and leather bite-resistant gloves for the protection of both personnel and the bats and were regularly tested for COVID-19 prior to work with live bats.

When mist netted, bats were immediately removed from the netting and held in cotton drawstring bags until processing (age, sex, and reproductive status determination, weight and length measurements, and clinical sample collection). Age classes were assigned as described previously [21]. Rectal and oral swabs in RNAlater (Thermo Fisher, Waltham, MA, USA), whole blood, wing punch biopsies, and any ectoparasites were collected and held on frozen cold packs before we released each bat where they were initially trapped. Samples were transferred to a −80 °C freezer at the end of each bout of collection.

Seven bats (three *Eptesicus fuscus*, three *Myotis* spp., and one *Lasiurus cinereus*) required humane euthanasia during handling because of observed injuries or abnormal neurologic behavior. Carcasses were frozen at −20 °C and shipped to the Wisconsin Veterinary Diagnostic Laboratory, where necropsies were performed in a biosafety cabinet. Tissue samples (heart, lung, liver, kidneys, spleen, small intestine, and colon) were collected and placed in RNAlater for molecular diagnostics.

**Table 1 viruses-17-01349-t001:** North American bats sampled as part of this study by species, sex, age class, and key behavior traits.

Species ^1^						Age Classes				
(By Sex)	Migrates	Peridomestic	Group Size	Sexes Cohabit Roost	Synchronized Births	Unknown	Adult	Subadult	Juvenile	Total
*Antrozous pallidus* [22,23]	Yes	Moderate	Few to >100	Yes	Yes	1	31	3		35
Female							25	3		28
Male							6			6
Not recorded						1				1
*Corynorhinus townsendii* [24,25,26]	Yes	Low	40-100	Yes	Yes		9			9
Female							8			8
Male							1			1
*Eptesicus fuscus* [27,28,29,30,31]	Yes	High	>100	Yes	Yes		11	2		13
Female							6	2		8
Male							5			5
*Lasiurus cinereus* [32,33]	Yes	Low	1	No	No		3			3
Female										0
Male							3			3
*Myotis* species ^2^ [29,34,35,36]	Yes	Moderate	>100	Varies	Yes		73	35	11	119
Female							48	16	4	68
Male							25	19	7	51
*Tadarida brasiliensis* [37,38,39,40]	Mix	Moderate	>1000	Yes	Yes	2	60	13		75
Female							36	4		40
Male							24	9		33
Not recorded						2				2
Total						3	187	53	11	254

^1^ All species in family Vespertilionidae except *T. brasiliensis* (Molossidae). References are for behavior traits listed for each species. ^2^ Includes *M. californicus*, *M. ciliolabrum*, *M. evotis*, *M. volans*, *M. yumanensis*, and uncharacterized *Myotis* species.

### 2.2. RNA Extraction

Total viral RNA was extracted from rectal swabs and colon biopsies using previously described methods [41,42,43]. In brief, Dacron swab tips were homogenized with 150 μL RNAlater and 850 μL Hanks’ Balanced Salt Solution (HBSS) or 10 mg tissue samples were homogenized with 600 μL HBSS, and samples were centrifuged to clarify. The supernatant was treated with nucleases to digest nucleic acids unencapsidated within viral particles [44]. Nucleic acids were then extracted using the QIamp MinElute Virus Spin Kit (Qiagen, Hilden, Germany), following the manufacturer’s protocol but omitting carrier RNA.

### 2.3. Pan-Coronavirus RT-PCR

Semi-nested reverse transcription PCR (RT-PCR) external reactions were performed using 0.4 μM degenerate primers (pan-CoV_outF: 5′- CCAARTTYTAYGGHGGITGG-3′ and pan-CoV_R: 5′- TGTTGIGARCARAAYTCATGIGG-3′), which are broadly reactive consensus primers targeting the ORF1ab RNA-dependent RNA polymerase (RdRp) of all four known genera of coronaviruses [45] and have been used successfully to detect coronaviruses in bat fecal samples [46,47]. RNA from three coronavirus cell culture isolates (transmissible gastroenteritis virus [TGEV], bovine coronavirus [BCoV], and infectious bronchitis virus [IBV]) and one synthetic RNA standard (SARS-CoV-2) (AcroMetrix COVID-19 RNA Control, Thermo Fisher) were included as positive controls. The limit of detection for this assay was between 20 and 100 viral copies per reaction, as determined by serial dilution using the quantitated SARS-CoV-2 standard (Appendix A).

The external RT-PCR was carried out in 20 μL reactions using the Qiagen OneStep RT-PCR kit and 5 μL of template RNA, according to manufacturer instructions, on a C-1000 thermocycler (BioRad, Hercules, CA, USA) with the following cycling conditions: 50 °C for 30 min; 95 °C for 15 min; 35 cycles of 94 °C for 40 s, 53.4 °C for 40 s, 72 °C for 1 min; and 72 °C for 10 min. Internal PCR reactions were carried out in 25 μL reactions containing 0.2 μM of each primer (pan-CoV_inF: 5′- GGTTGGGAYTAYCCHAARTGTGA-3′ and pan-CoV_R), 12.5 μL Qiagen 2x HotStarTaq DNA Polymerase, and 1 μL of external PCR product with the following cycling conditions: 95 °C for 15 min; 35 cycles of 94 °C for 30 s, 54.5 °C for 30 s, 72 °C for 1 min; and 72 °C for 10 min.

Internal PCR products were electrophoresed on a 2% agarose gel with ethidium bromide and 1 kb plus DNA length standards (New England Biolabs, Ipswich, MA, USA), visualized under UV light to confirm expected band lengths of approximately 600 base pairs (bp), and photographed using a GelDoc XR imager (BioRad). All positive PCR products were excised from gels and purified with the Zymoclean Gel DNA Recovery Kit (Zymo Research, Irvine, CA, USA), eluted in 6 μL provided elution buffer, and directly sequenced on both strands by using an ABI 3130xl Genetic Analyzer sequencer (Applied Biosystems, Foster City, CA, USA) at the University of Wisconsin-Madison Biotechnology Center for confirmation and characterization. We proofread and assembled chromatograms using Sequencher 4.10.1 (Gene Codes, Ann Arbor, MI, USA).

### 2.4. Metagenomic Sequencing

We performed metagenomic sequencing on a representative subset of 14 samples (eight rectal swabs and six colon biopsies), each from different animals from a diverse range of species and age–sex classes. Extracted RNA was subjected to reverse transcription with the Superscript IV kit (Thermo Fisher) using random hexamer priming. Resulting cDNA was then purified using Agencourt AMPure XP beads (Beckman Coulter, Brea, CA, USA) as previously described [43,48,49,50]. Genomic libraries were prepared using the Illumina Nextera XT kit (Illumina, San Diego, CA, USA) and sequenced on an Illumina MiSeq instrument using 300 × 300 cycle paired-end (V3) chemistry.

### 2.5. Bioinformatics

Sequences of low quality (Phred score <30) and short length (<50 bp) were trimmed and sequences matching known contaminants and host DNA were discarded using CLC Genomics Workbench v. 20.0.4 (Qiagen). Remaining reads were then subjected to de novo assembly using the metaviral command in SPAdes v. 3.15.2 [51]. The generated contiguous sequences (contigs) were compared to viruses in the NCBI GenBank database at both the nucleotide and amino acid levels using the BLASTn and BLASTx algorithms, respectively [52,53]. Contigs with high similarity to mammalian viruses were retained for further analysis whereas contigs matching viruses of non-mammalian hosts (e.g., phage, fungi, insects, or plants) were removed.

### 2.6. Phylogenetic Analysis

Phylogenetic relationships among related adenoviruses were inferred from nucleotide sequences available in NCBI GenBank. The hexon, penton, and polymerase genes of the novel adenovirus and closely related adenoviruses were aligned individually using MAFFT [54]. Positions with ambiguous residues were removed and the resulting alignments were trimmed with trimAl [55]. The aligned, trimmed sequences were then concatenated by hand. A maximum likelihood phylogenetic tree was inferred from the concatenated sequences (18 taxa, 1632 amino acid positions, and 772 variable positions) using IQ-TREE with the model of molecular evolution estimated from the data and 1000 ultrafast bootstrap replicates [56]. The resulting tree was displayed (outgroup rooted with bovine atadenovirus D) using FigTree 1.4.4 [57].

## 3. Results

### 3.1. Coronavirus Surveillance Yielded No Detections

All positive controls (TGEV, BCoV, IBV, and SARS-CoV-2) amplified with the pan-coronavirus RT-PCR. However, no coronavirus nucleic acids were detected in either rectal swabs or colon samples from 254 bats by RT-PCR or metagenomic sequencing (further described below).

### 3.2. Novel Virus Identification and Genomic Characterization

Following quality trimming and in silico subtraction of host and known contaminant sequences, we retained a total of 23,998,919 sequence reads with an average length of 873 bp for analysis. We successfully assembled one complete adeno-associated virus (AAV) genome of 4298 bp and partial coding sequences of an adenovirus (AdV), astrovirus (AstV), and genomovirus (GV) (Table 2) from rectal swab samples. All assembled sequences and raw reads have been deposited into NCBI GenBank (accession numbers PV983328-PV983331) and the NCBI Sequence Read Archive (accession numbers SRX29156395- SRX29156397), respectively.

The AdV was detected in an adult female big brown bat (*Eptesicus fuscus*), which was likely a geriatric individual due to significant dental wear [21]. Both the AAV and AstV were detected in one juvenile male *Myotis* species. The GV was detected in a juvenile female *Myotis* species. All three virus-positive bats were captured in urban environments near Las Cruces, New Mexico, either under a bridge or on a university campus.

The AdV polymerase gene was 70.9% similar at the amino acid level to the closest related virus (canine AdV 2, Genbank accession no. AP_000613). Because this AdV was detected in a new host species and has over 15% dissimilarity in the polymerase amino acid sequence to its closest relative, it is eligible for new species demarcation per ICTV guidelines [12]. Therefore, we have proposed the name lacepfus virus (LCPV), due to its detection in Las Cruces, New Mexico, USA, in a big brown bat (*Eptesicus fuscus*). LCPV is genetically more similar to canine adenoviruses than other published adenoviruses detected in bats to date (Figure 2).

## 4. Discussion

In this study, we collected rectal swabs from 254 bats in New Mexico, USA, from August 2020 to July 2021. We performed multimodal viral detection, including a pan-coronavirus RT-PCR on all samples and metagenomic sequencing on a subset of 14 samples. We detected no coronaviruses, but we identified four novel viruses, including an AdV, AAV, AstV, and GV, in three individuals. All four viruses were detected in rectal swabs and represented a diversity of genome structures (DNA and RNA, both single- and double-stranded).

The AdV (*Adenoviridae*), for which we propose the name lacepfus virus (LCPV), is more closely related to canine adenoviruses than to any other adenoviruses detected in bats to date. The first AdV detected in a bat was from a fruit bat (*Pteropus dasymallus yayeyamae*) [58]. Soon thereafter, researchers postulated that all canine AdVs originated from an AdV of a vespertilionid bat (such as *Eptesicus fuscus*, the species in which LCPV was detected), either through direct or indirect contact with a canid [59,60]. First discovered in 1930, canine AdV1 and AdV2, which cause hepatitis or tracheobronchitis (“kennel cough”), respectively, are significant pathogens in dogs, with CAdV1 causing mortality rates of 10–30% in unvaccinated animals [61]. Thus, both viruses are included in dog core vaccination series in the U.S. [62]. Although most AdV are host-specific, canine AdVs have been detected in a wide variety of carnivores (Figure 2) and are more severely pathogenic than the majority of AdVs, supporting the hypothesis that a historical interspecies transmission event, possibly from a bat, gave rise to these viruses [59].

AAVs (*Parvoviridae*) are widespread among vertebrates, including bats, and AAVs isolated from bats exhibit low sequence similarity to primate AAVs [63]. Similarly, AstVs (*Astroviridae*) have been detected extensively in a wide range of vertebrate hosts, often causing gastroenteritis and diarrhea in young individuals, but are considered zoonotic [64]. GVs (*Genomoviridae*) have been identified in animal, plant, and environmental samples, and are thought to be passed in bat guano from dietary sources [8,65].

None of the viruses detected in this study are definitively pathogenic in bats. Because these viruses were detected in antemortem rectal swabs, it was not possible to evaluate tropism in bat tissues. Histopathological confirmation of true infection would be useful in future studies. However, a recent study did perform histopathological characterization, providing evidence of AdV-induced enteritis in a juvenile Seba’s short-tailed bat (*Carollia perspicillata*) in Mexico [66]. Another previous study did not detect any viral or bacterial agents except for a novel AdV in three bats that died of “natural causes”, suggesting the AdV (Figure 2, bat adenovirus 2) may have been the etiologic agent [60]. Each of the virus-positive bats in our study appeared clinically healthy at the time of sample collection. Of note, all virus-positive bats were either juvenile or geriatric, supporting the hypothesis that viral shedding increases in bats, as in other species, during periods of immune system immaturity or senescence [67].

Of note, we did not detect coronavirus nucleic acids in any of the bats. Two recent studies of coronaviruses in bats from Canada [68] and the Yucatán region of Mexico [69] reported CoV prevalences of 1.4% and 5.4%, respectively. Other previous studies have detected enzootic alphacoronaviruses in native bat populations in the U.S. [4,6,7,8,9], Mexico [5], and Canada [70,71]. Our study relied on rectal swab samples, which have yielded the most, and often the only, detections of any sample type in previous studies [4,5,6,70], though it is possible other sample types could have yielded CoV detections. However, we hypothesize that coronavirus prevalence in these bats may have been relatively low due to the very hot, arid climate of our sampling location in the Chihuahua Desert; CoV virions are more stable and readily transmitted in cool, dry conditions [72,73]. If CoV prevalence was 1% in our study population, the probability of not detecting an infected bat (false negative) would have been relatively high at 8% [74]. Moreover, despite our efforts to sample a diverse range of species and age–sex classes frequently through all seasons, it is also possible we did not capture instances of peak shedding for coronaviruses in these species. Viral shedding in bats is known to be affected by seasonal fluctuations in immunity due to factors such as pregnancy, parturition, lactation, weaning, roosting behaviors, and hibernation [67,75,76,77].

The absence of SARS-CoV-2 in our sampled population is not surprising, given similar findings in other recent studies [78,79]. After experimental challenges with SARS-CoV-2, big brown bats (*Eptesicus fuscus*) and little brown bats (*Myotis lucifugus*) were resistant to infection [80,81], whereas Old World fruit bats (*Rosettus aegypticus*) could be infected readily and shed for several days post-inoculation [82]. Experimentally challenged Mexican free-tailed bats (*Tadarida brasiliensis*) shed SARS-CoV-2 in saliva for up to 18 days after inoculation but cleared the virus by day 20 [83]. However, none of the virus-positive animals infected co-housed bats, further suggesting that host competence and transmission risk of SARS-CoV-2 in this species remains low [83]. Notably, all bat samples for our study were collected before the emergence of the SARS-CoV-2 Omicron variant in October 2021, which was associated with significantly higher human-to-human transmission rates and thus viral prevalence compared to previous variants [84]. It is possible that increased transmission among humans could have allowed for more opportunities for SARS-CoV-2 to spill back into native U.S. bat populations after the conclusion of our study. To date, other mammalian species have been susceptible to reverse zoonotic transmission of SARS-CoV-2, and even documented transmission back to humans, including free-ranging white-tailed deer (*Odocoileus virginianus*) [85,86] and farmed mink (*Neovison vison*) [87].

Detection of LCPV opens new questions about bat–canid viral exchange that should be explored in future viral surveillance studies, especially studies of juvenile and geriatric bats that may yield higher viral detection rates due to immune system vulnerabilities. Bats live in peridomestic habitats, such as the urban sampling locations for this study. In fact, big brown bats and *Myotis* species, in which novel viruses were detected, are known to be moderately to highly synanthropic, roosting in groups of hundreds of individuals near humans and domestic animals. Dogs live with people and are susceptible to exposure to bat-borne viruses through bites or indirect contact (i.e., sniffing or eating guano). Transmission of histoplasmosis from bats to dogs in the same general area as this study has been documented [88]. It is unclear when the apparent AdV transmission event from bats to dogs occurred, or whether there were intermediary species involved, but such an event underscores the need to conduct viral surveillance in a broad range of animals that live with and near humans to understand bat-borne virus ecology and to evaluate the risk of future spillover.

## Figures and Tables

**Figure 1 viruses-17-01349-f001:**
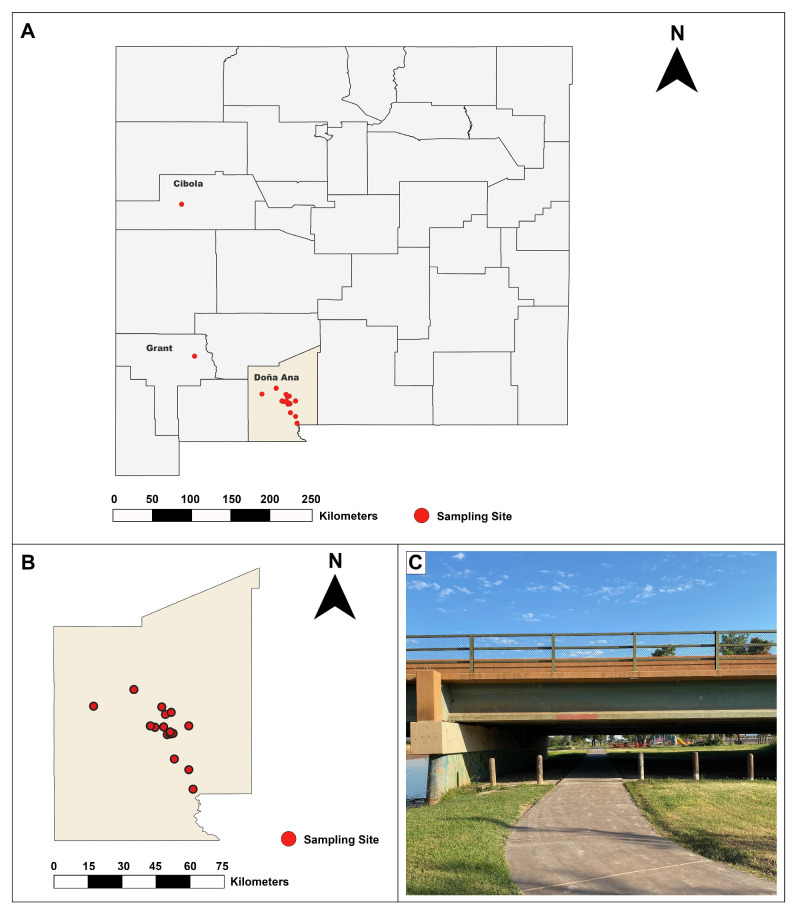
Bat sampling sites in New Mexico, USA: (**A**) Map of New Mexico counties, with sampling sites indicated as red dots. Counties in which sampling took place are labeled. (**B**) Map sampling sites in Doña Ana County, the most densely sampled county. (**C**) Representative photo of bridge sampling site in La Llorona Park, Doña Ana County.

**Figure 2 viruses-17-01349-f002:**
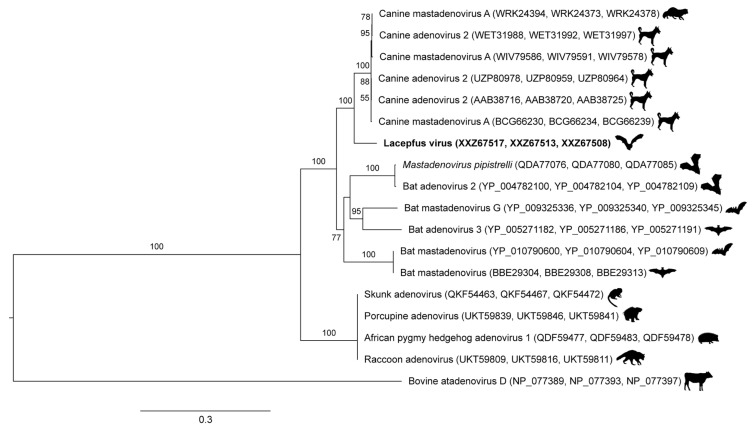
Maximum likelihood phylogenetic tree of adenoviruses based on aligned, concatenated hexon, penton and polymerase amino acid sequences. The tree is outgroup rooted with bovine atadenovirus D. Taxon names are followed by GenBank accession numbers in parentheses and silhouettes of the host of each virus. The taxon name of the virus identified in this study is in bold. Numbers beside nodes indicate bootstrap values (percent; only values ≥50% are shown); scale bar indicates nucleotide substitutions per site. For ease of reference, virus names reflect those listed in GenBank. Host silhouettes accessed from PhyloPic (phylopic.org) and used under Creative Commons licenses.

**Table 2 viruses-17-01349-t002:** Viruses identified in rectal swabs of free-ranging bats (n = 254) in New Mexico, USA, in August 2020–July 2021.

Host Species	Age/Sex	Virus Name	Accession	Genome	Sequence Length (nt) ^1^	Closest Match (Source, Location, Year, Accession)	% nt Similarity
*Eptesicus fuscus*	Geriatric female	Lacepfus virus (LCPV)	PV983329	dsDNA	31,279	Canine mastadenovirus A (feces, Turkey, 2022, OQ596341)	87.02
*Myotis* sp.	Juvenile male	Bat adeno-associated virus 2259	PV983328	dsDNA	4298	Bat adeno-associated virus YNM (rectal swab, China, 2008, GU226971)	77.09
*Myotis* sp.	Juvenile male	Bat astrovirus 2259	PV983331	ssRNA(+)	951	Bat astrovirus BAstV/RB (guano, USA, 2020, MT734809)	96.21
*Myotis* sp.	Juvenile female	Bat genomovirus 2252	PV983330	ssDNA	1303	Chicken genomovirus mg4_1247 (tracheal swab, USA, 2017, MN379609)	95.79

^1^ nt = nucleotide. All NCBI E-values = 0.00E+00.

## Data Availability

The original data presented in the study are openly available in NCBI GenBank (accession numbers PV983328-PV983331).

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
