# Peer review of "Novel Bat Adenovirus Closely Related to Canine Adenoviruses Identified via Fecal Virome Surveillance of Bats in New Mexico, USA, 2020–2021"

_viruses, 2025, doi:10.3390/v17101349_

Round 1
Reviewer 1 Report
Comments and Suggestions for Authors
Dear Authors!
New Article describes the bat monitoring data in new territory for almost year. The data is relevant and interesting. The any coronaviruses absence in material is of interest. New adenovirus increases genetic data (sequences) on this genus viruses diversity.
The Article combines different approaches to data obtaining, such as PCR diagnostics and metagenomic sequencing.
Comments and recommendations:
- Raw metagenomic sequencing reads are recommended deposited in the NCBI Sequence Read Archive (SRA) under access number. Specify the number in the article.
- Table 1. What information do references 21-39 confirm? This references are superfluous.
-Discussion describes very little of Table 1 data. For example, it was possible to expand of synanthropy (peridomestic) discussion.
- Add the length of fragment to Table 2.
- Add phylogenetic trees for PV983328, PV983331, PV983330 to the "Supplementary Materials".
Author Response
- Raw metagenomic sequencing reads are recommended deposited in the NCBI Sequence Read Archive (SRA) under access number. Specify the number in the article.
Thanks for highlighting this. The raw reads had been deposited previously in SRA, and the accession numbers are now included in the Results section (lines 224-225).
Table 1. What information do references 21-39 confirm? This references are superfluous.
The reviewer brings up a good question. These are references for the behavior traits listed in the columns of Table 1 for each species. We have now included a statement for clarification in the Table 1 footnote (line 117).
Discussion describes very little of Table 1 data. For example, it was possible to expand of synanthropy (peridomestic) discussion.
The reviewer makes an excellent point, especially because the adenovirus was detected in a bat species (Eptesicus fuscus) that is highly peridomestic. A section has been included in the Discussion elaborating on behavior information included in Table 1 (lines 333-337).
Add the length of fragment to Table 2.
The sequence length for each genome or partial coding sequence has been included in Table 2.
Add phylogenetic trees for PV983328, PV983331, PV983330 to the "Supplementary Materials".
The phylogenetic trees have been added to the Supplementary Materials (Figures S2-S4).
Reviewer 2 Report
Comments and Suggestions for Authors
The study “Novel bat adenovirus closely related to canine adenoviruses identified via fecal virome surveillance of North American bats, 2020-2021” focused on virological surveillance of bats of various species in urban, suburban, and rural settings in New Mexico, USA. A study screening bats in the southwestern U.S. for coronaviruses employed a broad-range RT-PCR test and metagenomic sequencing on samples. Despite this thorough analysis, no coronaviruses were detected in the bat population during the investigation. The research did, however, lead to the identification of four previously unknown viruses. A significant discovery was a new bat adenovirus showing a close genetic relationship to adenoviruses found in canines. These results suggest a lack of widespread coronavirus transmission among these bats but emphasize the importance of ongoing viral surveillance to monitor the potential for cross-species transmission between wildlife, domestic animals, and humans.
The paper is well written and I assume that this study may be published in the "Viruses" after the text has been corrected.
Comments:
In Abstract: four novel viruses were identified: an adenovirus (lacepfus virus, LCPV)…
The reader may not understand what the lacepfus virus is from the abstract. Only from the text of the manuscript can one understand that this name is proposed by the study's authors. However, the names of new viruses are approved by ICTV; until then, it is permissible to write about a "proposed" name for a new virus. This also applies to the text of the manuscript.
In Results: Because this AdV was detected in a new host species, it is eligible for new species demarcation per ICTV guidelines [12]. Therefore, we have named it lacepfus virus (LCPV)...
- The authors claim that bats are hosts for LCPV. However, this virus was identified in feces. Feces also contain bacteriophages, insect viruses, and other viruses whose hosts are not bats. Therefore, the authors can only speculate that bats are hosts for LCPV. A detailed analysis of the animal's organs is required for rigorous confirmation.
- The authors claim that LCPV is a new species of virus. However, phylogenetic analysis alone is insufficient to support this assertion. It is necessary to at least determine the genetic distances between different strains of the same adenovirus species and between different adenovirus species to assess whether distances exist at the species level.
Author Response
In Abstract: four novel viruses were identified: an adenovirus (lacepfus virus, LCPV)…
- The reader may not understand what the lacepfus virus is from the abstract. Only from the text of the manuscript can one understand that this name is proposed by the study's authors. However, the names of new viruses are approved by ICTV; until then, it is permissible to write about a "proposed" name for a new virus. This also applies to the text of the manuscript.
The reviewer brings up a good point. We have modified the text in the Abstract (line 26), Results (line 241), and Discussion (line 266) to reflect that lacepfus virus is a proposed name.
In Results: Because this AdV was detected in a new host species, it is eligible for new species demarcation per ICTV guidelines [12]. Therefore, we have named it lacepfus virus (LCPV)...
- The authors claim that bats are hosts for LCPV. However, this virus was identified in feces. Feces also contain bacteriophages, insect viruses, and other viruses whose hosts are not bats. Therefore, the authors can only speculate that bats are hosts for LCPV. A detailed analysis of the animal's organs is required for rigorous confirmation.
We thank the reviewer for this thoughtful comment. Given that mastadenoviruses, such as LCPV, have only been detected in mammals, it is highly unlikely that the source was another host present in the feces or environment. However, the reviewer is correct that analysis of tissue tropism would be useful for confirming bat infection. We have included a statement to this effect in the Discussion (lines 285-288).
- The authors claim that LCPV is a new species of virus. However, phylogenetic analysis alone is insufficient to support this assertion. It is necessary to at least determine the genetic distances between different strains of the same adenovirus species and between different adenovirus species to assess whether distances exist at the species level.
The reviewer is indeed correct. ICTV criteria for a new species of mastadenovirus include >10-15% phylogenetic distance of the polymerase amino acid sequence and novel host range, both of which are satisfied in the case of LCPV. We agree that listing these species demarcation criteria in the body of the text would be helpful and have included this information in the Results (lines 238-239).
Reviewer 3 Report
Comments and Suggestions for Authors
Weary et al. conducted viral surveillance on fecal samples from 10 bat species (254 individuals in total) in New Mexico, North America, from 2020 to 2021. Combining pan-coronavirus RT-PCR and metagenomic sequencing techniques, no coronaviruses (including SARS-CoV-2) were detected; however, four novel viruses were identified: an adenovirus (Lacepfus virus, LCPV), an adeno-associated virus (AAV), an astrovirus (AstV), and a genomovirus (GV). Among these, LCPV exhibited higher genetic similarity to canine adenoviruses than to other bat adenoviruses, suggesting potential historical viral transmission between bats and dogs. The study design focused on the "bat-human-other animals" viral transmission chain, filling the gap in research on the bat virome in arid regions of North America and providing basic data for assessing the risk of cross-species viral transmission. Before publication, the manuscript needs further refinement to enhance academic rigor and application value.
1.Bats harbor a large number of known and unknown pathogens. The study collected a large number of bat samples, and researchers had frequent direct contact with bats. Has consideration been given to the possibility that bat viruses may gradually evolve through contact with humans and eventually cross species to infect humans?
2.Although the title "Novel bat adenovirus closely related to canine adenoviruses identified via fecal virome surveillance of North American bats, 2020-2021" clearly highlights the core finding, the geographical scope described by "North American bats" is overly broad. In fact, the study only targeted bats in New Mexico and did not cover other regions of North America, which may mislead readers into thinking that the research is representative of the entire North American continent.
3.The Introduction only mentions that "bat adenovirus research is relatively limited" and fails to integrate key studies conducted after 2020 (e.g., the evolutionary relationship between a novel adenovirus identified in bats from Kazakhstan by Karamendin et al. in 2023 and canine adenoviruses).
4.The manuscript only mentions a 70.9% amino acid similarity in the polymerase gene with canine adenovirus type 2, but does not cite the International Committee on Taxonomy of Viruses (ICTV) classification criteria for the Adenoviridae family, making it impossible to clarify the basis for classifying LCPV as a "new species".
5.In accordance with the latest ICTV classification, it is recommended to uniformly label "Canine adenovirus 2" in the phylogenetic tree as "Canine mastadenovirus A".
6.Figure 2 (the maximum likelihood phylogenetic tree) does not indicate the genetic distance between LCPV and canine adenoviruses (e.g., the amino acid substitution rate corresponding to branch lengths), and the "host silhouettes" lack a consistent style (some species have no silhouettes), which affects readability and requires revision.
Author Response
1.Bats harbor a large number of known and unknown pathogens. The study collected a large number of bat samples, and researchers had frequent direct contact with bats. Has consideration been given to the possibility that bat viruses may gradually evolve through contact with humans and eventually cross species to infect humans?
Many thanks for this comment. During sampling for this study, each bat was only caught and sampled once. Also, for human and bat safety, and in keeping with the exemption we were granted to conduct bat research during the early COVID-19 embargo period, all bat handlers wore N95 masks, face shields, and leather bite-resistant gloves. We have included a statement describing these biosecurity precautions in the Methods (lines 96-100).
2.Although the title "Novel bat adenovirus closely related to canine adenoviruses identified via fecal virome surveillance of North American bats, 2020-2021" clearly highlights the core finding, the geographical scope described by "North American bats" is overly broad. In fact, the study only targeted bats in New Mexico and did not cover other regions of North America, which may mislead readers into thinking that the research is representative of the entire North American continent.
We thank the reviewer for their close reading of all parts of the manuscript, including the title. “North American bats” was intended to describe the sampled bat species in that bats of these species can be found throughout North America. We agree that this may be confusing to readers and have changed the title to read “…surveillance of bats in New Mexico, USA, 2020-2021.”
3.The Introduction only mentions that "bat adenovirus research is relatively limited" and fails to integrate key studies conducted after 2020 (e.g., the evolutionary relationship between a novel adenovirus identified in bats from Kazakhstan by Karamendin et al. in 2023 and canine adenoviruses).
We thank the reviewer for this critique. In addition to Karamendin et al. (2023), we also cite and incorporate information from other bat adenovirus studies published in the past five years from diverse geographic areas, such as Ntumvi et al. (2021) in Sub-Saharan Africa and Dias et al. (2024) in South America, although AdV research remains limited compared to other bat-borne viruses, such as rabies and coronaviruses. Recently detected bat AdVs were omitted from our phylogenetic tree for the sake of clarity if they were much less similar to canine AdV strains than LCPV, such as the virus characterized in Karamendin et al., or too similar to previously described bat AdVs as to be redundant. We have, however, conducted a comprehensive literature review and discovered a 2025 paper by Trejo-Chavez et al. that lends anatomic and histologic evidence of AdV enteric pathology in a New World bat. We have included discussion of their findings (lines 288-290).
4.The manuscript only mentions a 70.9% amino acid similarity in the polymerase gene with canine adenovirus type 2, but does not cite the International Committee on Taxonomy of Viruses (ICTV) classification criteria for the Adenoviridae family, making it impossible to clarify the basis for classifying LCPV as a "new species".
The reviewer is indeed correct. ICTV criteria for a new species of mastadenovirus include >10-15% phylogenetic distance of the polymerase amino acid sequence and novel host range, both of which are satisfied in the case of LCPV. Although already cited (reference 12), we agree that listing these species demarcation criteria in the body of the text would be helpful and have included a section in the Results (lines 238-239).
5.In accordance with the latest ICTV classification, it is recommended to uniformly label "Canine adenovirus 2" in the phylogenetic tree as "Canine mastadenovirus A".
We are grateful to the reviewer for their careful consideration of our figures. For ease of reference, we have used the nomenclature used in the GenBank entries for each of these viruses. We have added this explanation to the figure caption (lines 254-255).
6.Figure 2 (the maximum likelihood phylogenetic tree) does not indicate the genetic distance between LCPV and canine adenoviruses (e.g., the amino acid substitution rate corresponding to branch lengths), and the "host silhouettes" lack a consistent style (some species have no silhouettes), which affects readability and requires revision.
The nucleotide substitution rate is displayed at the bottom of the tree (0.3 nucleotide substitutions per site) and explained in the figure caption (lines 253-254). We have carefully reviewed the host silhouettes and have confirmed that each branch has a corresponding icon. All icons are standardized and were taken from the PhyloPic database (phylopic.org). In this review, however, we realized we overlooked citing PhyloPic and have added this information to the Figure 2 caption (lines 255-256).
Round 2
Reviewer 2 Report
Comments and Suggestions for Authors
The study “Novel bat adenovirus closely related to canine adenoviruses identified via fecal virome surveillance of North American bats, 2020-2021” focused on virological surveillance of bats of various species in urban, suburban, and rural settings in New Mexico, USA. A study screening bats in the southwestern U.S. for coronaviruses employed a broad-range RT-PCR test and metagenomic sequencing on samples. Despite this thorough analysis, no coronaviruses were detected in the bat population during the investigation. The research did, however, lead to the identification of four previously unknown viruses. A significant discovery was a new bat adenovirus showing a close genetic relationship to adenoviruses found in canines. These results suggest a lack of widespread coronavirus transmission among these bats but emphasize the importance of ongoing viral surveillance to monitor the potential for cross-species transmission between wildlife, domestic animals, and humans.
The authors of the publication answered all the questions posed. The paper is well written and I assume that this study may be published in the "Viruses" .